

# Characterization of bacterioplankton communities and quantification of organic carbon pools off the Galapagos Archipelago under contrasting environmental conditions

Nataly Carolina Guevara Campoverde[1,2,*], Christiane Hassenrück[2,*], Pier Luigi Buttigieg[3] and Astrid Gärdes[2]

[1] Galapagos Science Center, Colegio de Ciencias Biológicas y Ambientales, Universidad San Francisco de Quito, Quito, Ecuador
[2] Tropical Marine Microbiology, Department of Biogeochemistry and Geology, Leibniz-Centre for Tropical Marine Reseach, Bremen, Germany
[3] HGF-MPG Group for Deep Sea Ecology and Technology, Alfred-Wegener-Institut, Helmholtz-Zentrum für Polar- und Meeresforschung, Bremerhaven, Germany
[*] These authors contributed equally to this work.

Corresponding author
Christiane Hassenrück,
christiane.hassenrueck@leibniz-zmt.de

## ABSTRACT

Bacteria play a crucial role in the marine carbon cycle, contributing to the production and degradation of organic carbon. Here, we investigated organic carbon pools, aggregate formation, and bacterioplankton communities in three contrasting oceanographic settings in the Galapagos Archipelago. We studied a submarine $CO_2$ vent at Roca Redonda (RoR), an upwelling site at Bolivar Channel (BoC) subjected to a weak El Niño event at the time of sampling in October 2014, as well as a site without volcanic or upwelling influence at Cowley Islet (CoI). We recorded physico-chemical parameters, and quantified particulate and dissolved organic carbon, transparent exopolymeric particles, and the potential of the water to form larger marine aggregates. Free-living and particle-attached bacterial communities were assessed via 16S rRNA gene sequencing. Both RoR and BoC exhibited temperatures elevated by 1–1.5 °C compared to CoI. RoR further experienced reduced pH between 6.8 and 7.4. We observed pronounced differences in organic carbon pools at each of the three sites, with highest dissolved organic carbon concentrations at BoC and RoR, and highest particulate organic carbon concentrations and aggregate formation at BoC. Bacterioplankton communities at BoC were dominated by opportunistic copiotrophic taxa, such as *Alteromonas* and *Roseobacter*, known to thrive in phytoplankton blooms, as opposed to oligotrophic taxa dominating at CoI, such as members of the SAR11 clade. Therefore, we propose that bacterial communities were mainly influenced by the availability of organic carbon at the investigated sites. Our study provides a comprehensive characterization of organic carbon pools and bacterioplankton communities, highlighting the high heterogeneity of various components of the marine carbon cycle around the Galapagos Archipelago.

## INTRODUCTION

Bacteria play a major role in the marine carbon cycle as primary producers, as well as organic matter degraders. In surface waters, photosynthetic microorganisms convert $CO_2$ into organic carbon, effectively reducing the $pCO_2$ concentration in the upper ocean. While the majority of the organic carbon produced by photosynthesis is respired by heterotrophic organisms and released back into the atmosphere as $CO_2$ (*Hutchins et al., 2017*), a small portion escapes remineralization and sinks to deeper waters as particulate organic carbon (POC) in the form of marine aggregates (*Alldredge & Silver, 1988*; *Ducklow et al., 2001*; *Giering et al., 2014*). The formation of these marine aggregates may be enhanced by the presence of transparent exopolymeric particles (TEP; *Passow, 2002*), which can be formed by cellular exudation from living or lysed cells (*Passow et al., 2001*; *Bhaskar et al., 2005*), and which, due to their sticky nature, function as biological glue facilitating aggregation (*Jackson, 1995*; *Engel, 2000*). Marine aggregates are heavily colonized by microorganisms and constitute a hotspot of bacterial activity since they contain inorganic and organic nutrients in abundance (*Azam & Malfatti, 2007*). Particle-attached (PA) bacteria are able to hydrolyze and resuspend the POC of marine aggregates (*Azam & Malfatti, 2007*; *Giering et al., 2014*), thereby reducing the vertical flux of organic matter (*Grossart & Ploug, 2001*). Conversely, free-living (FL) bacteria, which often constitute taxonomically and functionally distinct communities (*Rieck et al., 2015*), thrive on dissolved organic carbon (DOC) in the ocean (*Azam & Malfatti, 2007*). As such, the fate of organic carbon pools in the ocean is tightly coupled to the composition and activity of the FL and PA bacterioplankton community.

The Galapagos Archipelago consists of a group of volcanic islands, located in a complex system of tropical and temperate ocean currents, and intense upwelling zones. Seasonal upwelling occurs on the western side of the Galapagos Archipelago, where the Equatorial Undercurrent (EUC) collides with the iron-rich island platform and is diverted upwards, bringing colder (18 °C) and inorganic nutrient-rich waters to the surface (*Houvenaghel, 1978*; *Palacios, 2002*). The increased nutrient supply during the upwelling season is creating a productive area where massive phytoplankton blooms occur and large amounts of carbon are converted into organic matter (*Jimenez, 1981*; *Feldman, 1986*). Every few years, this seasonal upwelling is suppressed, and surface waters become warmer (26 °C) and depleted of their nutrients. This anomaly, known as El Niño, is thereby preventing the formation of phytoplankton blooms (*Houvenaghel, 1978*; *Sweet et al., 2007*; *Liu et al., 2014*). The eastern side of the Galapagos Archipelago is governed by the South Equatorial Current (SEC) and does not experience upwelling (*Schaeffer et al., 2008*). Here, although inorganic nutrient concentrations, specifically nitrate, may reach higher concentrations, primary production is generally limited by the availability of iron, resulting in conditions comparable to the high-nitrate low-chlorophyll (HNLC) areas that characterize most of the eastern and central equatorial Pacific in the absence of an upwelling system (*Schaeffer et al., 2008*). The Galapagos Archipelago is furthermore among the world's most active volcanic areas today. As a consequence, many islands are characterized by submarine gas seepage through fumaroles consisting mostly of $CO_2$ (*Standish et al., 1998*), thereby locally reducing seawater pH and altering the chemical composition of the seawater. Given these

contrasting environmental conditions throughout the Galapagos Archipelago, distinct FL and PA bacterioplankton communities as well as organic carbon pools and aggregation processes are expected to occur in different regions of the Galapagos Archipelago.

Previous studies documenting aspects of the marine carbon cycle off the Galapagos Archipelago focused mostly on upwelling and El Niño events, using satellite imagery in combination with *in situ* measurements to quantify physical water properties, inorganic nutrients and chlorophyll a as proxy for primary production (*Palacios, 2004*; *Sweet et al., 2007*; *Schaeffer et al., 2008*). Data on organic carbon pools around the Galapagos Archipelago are scarce and so far restricted to POC inferred from remote sensing information (*Gardner, Mishonov & Richardson, 2006*; *Kislik et al., 2017*). A direct quantification of DOC and POC, as well as TEP and aggregation processes, is still lacking for this region. Furthermore, apart from isolated observations during global ocean sampling campaigns (*Rusch et al., 2007*; *Yilmaz et al., 2012*), no studies have as yet reported on bacterioplankton communities and their potential role in the marine carbon cycle under the varied environmental settings of the Galapagos Archipelago.

The objective of the current study was to investigate various components of the marine carbon cycle by jointly exploring organic carbon pools, aggregate formation, and bacterial community composition under three contrasting environmental conditions around the Galapagos Archipelago. The study sites were selected based on expected temperature and pH differences: seasonal warming in an area otherwise influenced by upwelling, localized pH reduction, and an area outside the upwelling or pH reduction zones. We aimed: (i) to compare the physico-chemical conditions; (ii) to quantify changes in organic carbon pools (POC, DOC, TEP), including the potential for marine aggregate formation; and (iii) to assess differences in the diversity and composition of FL and PA bacterial communities in the water column among these sites. Thereby, we identified bacterial taxa that displayed strong changes in their contribution to bacterioplankton communities at each of the sites, discussing their importance in the marine carbon cycle under the selected environmental conditions.

## MATERIALS AND METHODS
### Study site
The sampling was conducted at three sites around the Galapagos Archipelago during a week-long cruise from October 20 to October 27, 2014 (research permit: PNG Permiso de investigacion científica PC-86-14; Fig. 1). The Bolivar Channel (BoC) west of Isabela Island (latitude: −0.2560667°, longitude: −91.41733611°) is located in an upwelling region, but was exhibiting elevated temperatures caused by the occurrence of a weak El Niño event during the sampling period. Roca Redonda (RoR), located about 25 km north of Isabela Island (latitude: 0.2817528°, longitude: −91.61177222°), is the nascent peak of a submarine volcano (*Standish et al., 1998*). $CO_2$ is released through numerous shallow (10–18 m) vents on the south-eastern side of the peak, locally acidifying the water column. Cowley Islet (CoI), located east of Isabela Island (latitude: −0.3861444°, longitude: −90.96356667°) showed similar bathymetry to RoR and BoC, while lacking detectable influences from upwelling or geothermal processes.

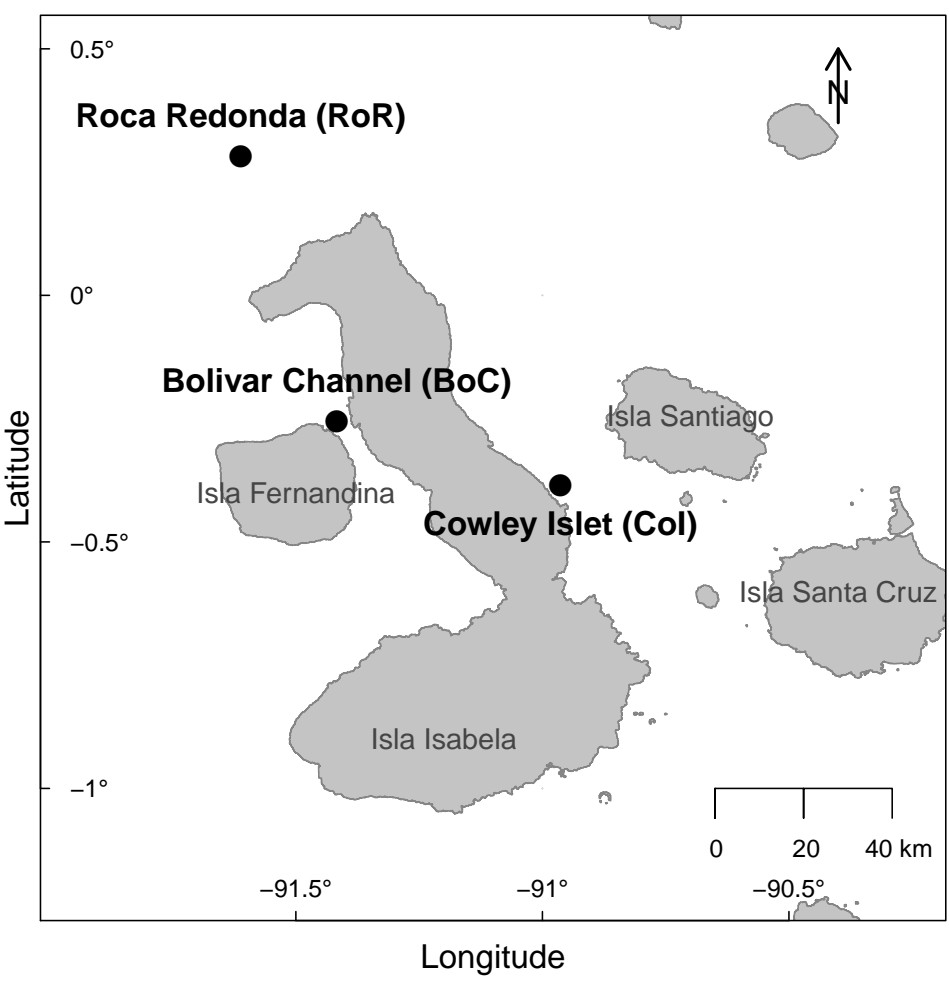

**Figure 1** Map of the sampling sites at Cowley Islet (CoI), Roca Redonda (RoR), and Bolivar Channel (BoC) around the Galapagos Islands.

## Water sampling

At each site, spatially separated ($\geq 20$ m) sub-sites with similar characteristics were selected to replicate the conditions of each site across independent water samples. At CoI and BoC five sub-sites were selected, while at RoR only four sub-sites were sampled due to the limited locations offering similar conditions at accessible depth. At each sub-site, seawater was collected at a depth of approximately 15 m using a 3 L Niskin bottle. At RoR this sampling depth corresponded to 50 cm above the vent. Seawater samples were exported to the Centre for Tropical Marine Research (ZMT), Bremen, Germany, with the approval of the Environmental Ministry of Ecuador (export clearance number 2419373).

Temperature, pH, salinity, dissolved oxygen and chlorophyll a concentration of the water column were measured with a Eureka MANTA 2 multi-probe (Eureka Environmental Engineering, Austin, TX, USA), at the same depth the seawater samples were taken. The concentrations of inorganic nutrients (nitrate/nitrite and phosphate) were measured using a continuous flow injection analyzer (FIAstar 5000, Foss Tecator, Hilleroed, Denmark).

To investigate different components of the organic carbon pool in the water column, we quantified DOC, TEP, and POC. DOC concentrations were measured by the high-temperature combustion (HTC) technique (*Wangersky, 1978*; *Dafner & Wangersky, 2002*), using a TOC-VCPH analyzer (Shimadzu, Mandel, Guelph, Canada). For the determination of TEP concentrations, 300 mL water samples were filtered through 0.45 μm polycarbonate filters at a constant vacuum of 0.160 bar. TEP was then measured semi-quantitatively using the Xanthan equivalent method (*Engel, 2009*; *Passow & Alldredge, 1995*). To quantify POC, we filtered 2 L of seawater through a pre-combusted (at 400 °C) and pre-weighed 47 mm glass microfiber filter (GF/F) in the field using a Millipore® all-glass filtration apparatus. After filtration, we stored the filters in plastic Petri dishes at room temperature (22 °C) until they were dried overnight at 40 °C in the laboratory and weighed again. We determined the concentration of POC per litre as the difference in filter weight divided by the filtered seawater volume. Data on measured water and carbon pool parameters are available on Pangaea (https://doi.pangaea.de/10.1594/PANGAEA.890858).

## Rolling tank experiments

To induce the natural formation of marine aggregates and simulate the continuous sinking of the particles, we performed rolling tank experiments (*Shanks & Edmondson, 1989*). One cylindrical Plexiglas rolling tank was filled with 1.5 L of the seawater collected at each sub-site. The tanks were rotated for 48 h at a constant speed of three turns per minute. Every 24 h, we measured aggregate formation by counting aggregates and classifying them into five size categories (<1 mm, 1–3 mm, 3–5 mm, 5–10 mm and >10 mm). The rotation of the tanks was briefly interrupted to count and measure the aggregates non-invasively. Based on the number of aggregates in each category, we calculated the total volume of aggregates per tank, assuming spherical shapes (*Cárdenas et al., 2015*). Data on aggregation potential are available on Pangaea (https://doi.pangaea.de/10.1594/PANGAEA.890858).

## Bacterial community composition

The FL and PA fractions of the bacterial community were separated via serial filtration. One L of seawater was first filtered through a 3.0 μm polycarbonate filter to collect the PA fraction, followed by filtration through a 0.2 μm polycarbonate filter to collect the FL fraction of the bacterial community. Two to four technical replicates were taken at each sub-site. The polycarbonate filters were folded into 1.5 ml Eppendorf tubes with 1 mL of RNAlater solution (Ambion) and immediately stored at −20 °C until DNA extraction. DNA was extracted individually from each technical replicate for both size fractions following the protocol developed by *Boström et al. (2004)* for low biomass samples. The DNA was quantified using a Bio Photometer Plus spectrophotometer (Eppendorf, Hamburg, Germany). To assess the diversity and composition of the bacterial community, the V3–V4 hypervariable region of the bacterial 16S rRNA gene was sequenced on the Illumina MiSeq PE platform at LGC Genomics GmbH (Berlin, Germany) using a 2 ×300 bp paired-end approach with universal bacterial primers (*Klindworth et al., 2013*). Separate libraries were prepared for technical replicates, which were then individually sequenced.

The primer-clipped sequences provided by LGC Genomics were further processed according to (*Hassenrück et al., 2016*). Sequences were quality trimmed with a sliding

window of four bases and a minimum average quality score of 15 using trimmomatic version 0.32 (*Bolger, Lohse & Usadel, 2014*). Paired-end reads were merged with PEAR version 0.9.5 (*Zhang et al., 2014*) using a minimum overlap of 10 base pairs, retaining only merged reads with a minimum and maximum sequence length of 350 and 500 base pairs, respectively. OTU clustering was performed with swarm version 2.0 using the fastidious algorithm with default parameters (*Mahé et al., 2014*). OTUs consisting of only one sequence (singletons) were removed from the data set. As representative sequence per OTU, the seed sequence of each swarm was taxonomically classified with silvangs with default settings (https://www.arb-silva.de/ngs/; date accessed 02.08.2017) using the SILVA ribosomal RNA gene database version 128 as reference (*Quast et al., 2013*). Only OTUs that occurred in at least two of the technical replicates per sub-site and size fraction were included in the final data set. OTUs of technical replicates were then merged into one community profile per sub-site by taking the sum of the sequence counts per OTU. Furthermore, OTUs with a sequence similarity of less than 93% to the reference database, OTUs unclassified at phylum level, and OTUs matching chloroplast and mitochondrial sequences were excluded from the analysis. The primer-clipped sequences were deposited in the European Nucleotide Archive (ENA, *Toribio et al., 2017*) using the data brokerage service of the German Federation for Biological Data (GFBio; *Diepenbroek et al., 2014*), in compliance with the Minimal Information about any (X) Sequence (MIxS) standard (*Yilmaz et al., 2011*) and are available with the accession PRJEB27168. The final OTU tables are accessible at Pangaea (https://doi.pangaea.de/10.1594/PANGAEA.890858).

## Statistical analysis

We performed a principal component analysis (PCA) on the measured water and carbon pool parameters after standardizing each variable using z-scoring (*Legendre & Legendre, 1998*; *Ramette, 2007*). Kruskal–Wallis tests were used to detect differences in water and carbon pool parameters between study sites, followed by pairwise comparisons based on Wilcoxon tests with False Discovery Rate (FDR)-corrected $p$-values (*Benjamini & Hochberg, 1995*). Aggregation potential was assessed based on the increase in aggregate volume over time during the rolling tank experiments. Differences in aggregate volume between the study sites were tested using general linear mixed models (GLMMs) with sub-site, i.e., rolling tank, as random factor.

Alpha diversity, specifically effective species richness, of the bacterial communities in the FL and PA fraction was assessed with the inverse Simpson index (*Chao et al., 2014*). No subsampling was required to account for unequal library sizes, since rarefaction curves based on the inverse Simpson index, unlike OTU number, were saturated at observed sequencing depths (Fig. S1). Differences in alpha diversity between FL and PA bacterial communities were tested with a paired Wilcoxon test. Patterns in alpha diversity between sampling sites were then analyzed separately within each size fraction using Kruskal–Wallis tests. $P$-values of pairwise comparisons based on Wilcoxon tests were adjusted using FDR correction (*Benjamini & Hochberg, 1995*). The relationship between alpha diversity and temperature and pH was assessed via Spearman rank correlations.

Differences in the composition of the bacterial communities (beta diversity) among samples were explored by cluster analysis and tested by analysis of similarity (ANOSIM) using a Bray–Curtis dissimilarity matrix calculated separately from the OTU data of the FL and PA fraction. Similarity percentages (SIMPER) were used to identify the OTUs contributing most to the dissimilarity between the FL and PA fraction within each water sample. The amount of variation in bacterial community composition explained by sampling site as well as temperature and pH, as parameters of specific interest, was assessed via redundancy analysis (RDA). Separate RDAs were performed for FL and PA communities. Prior to RDA, the data set was reduced by removing OTUs with low sample coverage and rare OTUs, i.e., OTUs occurring in less than 50% of the samples from at least one site and those contributing less than 1% to the sequences in at least one sample. There were four main reasons for this reduction step: (i) we were foremost interested in patterns among dominant OTUs. (ii) We wanted to remove statistically uninformative OTUs. OTUs with a low sample coverage are likely outliers and not representative of the conditions in the sampled area, and may therefore bias statistical tests. (iii) To perform further statistical analyses, a centered log ratio (clr)-transformation was applied to the OTU matrix. To enable the log operation a pseudo-count of 0.5 was added to all sequence counts in the OTU matrix (*Fernandes et al., 2014*). If an OTU matrix contains too many zeros, this addition may bias the original data. Therefore, rare OTUs were excluded to reduce the proportion of zero counts in the OTU matrix. (iv) By reducing the number of OTUs we were further able to increase the sensitivity of statistical approaches requiring multiple testing by applying a less severe $p$-value correction due to the reduced number of multiple tests. While this removal affected 99% and 96% of the OTUs, and 33% and 25% of the sequences, in the FL and PA fractions, respectively, we confirmed that it did not alter trends in beta diversity (Mantel test, $r > 0.99$, $p < 0.001$). Sequence counts were further clr-transformed with the aldex.clr function of the R package ALDEx2, using the median of 128 Monte Carlo Dirichlet instances (*Fernandes et al., 2014*). ALDEx2 was also used to identify differentially abundant OTUs between sampling sites in the reduced data set of each size fraction at a parametric FDR-adjusted and a non-parametric unadjusted significance threshold of 0.05.

All statistical analyses were conducted in R, using the R core distribution (*R Core Team, 2017*) with the additional R packages lmerTest (*Kuznetsova, Brockhoff & Christensen, 2016*), vegan (*Oksanen et al., 2017*), and ALDEx2 (*Fernandes et al., 2014*). All scripts (R and bash) for sequence processing and statistical data analysis are available with the data set on Pangaea (https://doi.pangaea.de/10.1594/PANGAEA.890858).

# RESULTS

## Water parameters and organic carbon pools

We observed pronounced differences in measured water parameters and the components of the organic carbon pool in the water column between the sampling sites at CoI, RoR and BoC. Both RoR and BoC exhibited temperatures elevated by approximately 1 to 1.5 °C compared to CoI, where values of about 22 °C were recorded (Table 1). Highest

**Table 1  Summary of observed water parameters and alpha diversity of free-living and particle-attached bacterial communities in the water column.**

| Parameter[a] | T [°C] | pH | Salinity [psu] | DO [mg L$^{-1}$] | NO$_x^-$ [μmol L$^{-1}$] | PO$_4^{3-}$ [μmol L$^{-1}$] |
|---|---|---|---|---|---|---|
| Cowley Islet (CoI) | | | | | | |
| Min | 21.35 | 8.14 | 34.67 | 6.65 | 4.65 | 0.55 |
| Median | 21.96 | 8.16 | 34.69 | 6.78 | 5.19 | 0.59 |
| Max | 22.13 | 8.25 | 34.72 | 6.83 | 5.80 | 0.68 |
| Group[b] | a | a | a | a | a | a |
| Roca Redonda (RoR) | | | | | | |
| Min | 22.14 | 6.83 | 34.43 | 6.23 | 2.60 | 0.43 |
| Median | 22.80 | 7.22 | 34.53 | 6.60 | 3.59 | 0.49 |
| Max | 23.32 | 7.35 | 34.64 | 8.19 | 4.49 | 0.52 |
| Group[b] | b | b | b | a,b | b | b |
| Bolivar Channel (BoC) | | | | | | |
| Min | 22.39 | 8.25 | 34.38 | 6.09 | 4.76 | 0.60 |
| Median | 23.24 | 8.32 | 34.62 | 6.43 | 5.12 | 0.64 |
| Max | 23.80 | 8.35 | 34.67 | 6.46 | 5.92 | 1.26 |
| Group[b] | b | c | b | b | a | a |

| Parameter[a] | Chl a [μg L$^{-1}$] | DOC [μmol L$^{-1}$] | TEP [μg X$_{eq}$ L$^{-1}$] | POC [mg L$^{-1}$] | InvS (FL) | InvS (PA) |
|---|---|---|---|---|---|---|
| Cowley Islet (CoI) | | | | | | |
| Min | 0.23 | 75.65 | 184.23 | 23.45 | 19.55 | 28.66 |
| Median | 0.35 | 80.87 | 235.40 | 32.93 | 33.82 | 48.12 |
| Max | 0.38 | 88.57 | 346.76 | 37.03 | 40.50 | 65.62 |
| Group[b] | a | a | ns | a | a | a |
| Roca Redonda (RoR) | | | | | | |
| Min | 0.09 | 88.96 | 142.57 | 18.46 | 2.31 | 4.41 |
| Median | 0.13 | 100.54 | 293.20 | 20.64 | 21.04 | 17.80 |
| Max | 0.19 | 112.73 | 323.67 | 22.37 | 24.99 | 20.29 |
| Group[b] | b | b | ns | b | a,b | b |
| Bolivar Channel (BoC) | | | | | | |
| Min | 0.15 | 86.76 | 243.33 | 25.38 | 4.19 | 3.16 |
| Median | 0.33 | 92.14 | 303.36 | 36.65 | 8.46 | 4.85 |
| Max | 0.40 | 97.99 | 459.05 | 44.09 | 14.75 | 7.67 |
| Group[b] | a | b | ns | a | b | b |

**Notes.**

[a]T, temperature; DO, dissolved oxygen; NO$_x^-$, nitrate/nitrite; PO$_4^{3-}$, phosphate; Chl a, chlorophyll a; DOC, dissolved organic carbon; TEP, transparent exopolymers; POC, particulate organic carbon; InvS (FL), Inverse Simpson index of free-living bacterial communities; InvS (PA), Inverse Simpson index of particle-attached bacterial communities.

[b]Lower case letters indicate membership to significantly different groups based on pairwise Wilcoxon posthoc tests between sampling sites (ns, no significant differences among sites based on Kruskal Wallis tests).

temperatures occurred at BoC with maximum values of 23.8 °C. pH ranged from 6.83 at RoR to 8.35 at BoC. Whereas at CoI and BoC measured pH values were always above 8, observed pH at RoR did not exceed 7.35 (Table 1). Salinity only varied within a small range among all three sites from 34.38 to 34.72 psu. With the exception of one oxygen measurement of 8.19 mg L$^{-1}$ at RoR, dissolved oxygen concentrations were highest at

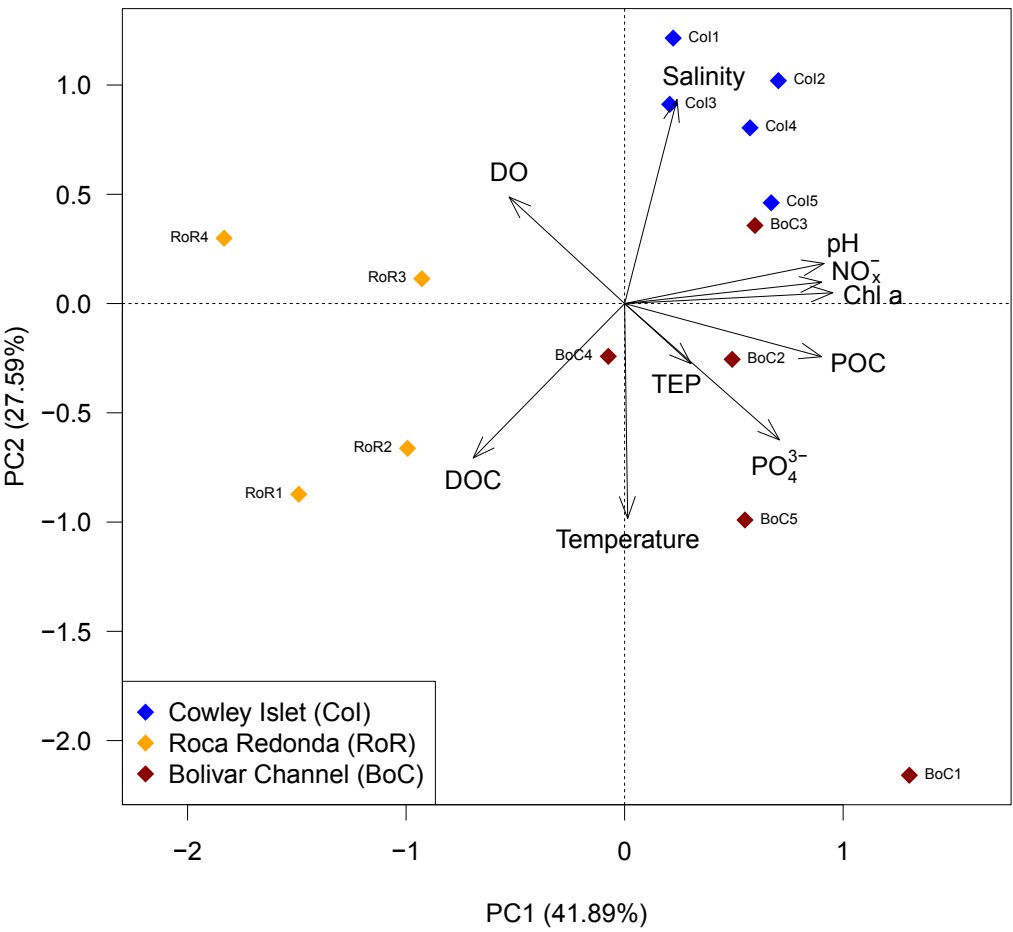

**Figure 2** **Principal component analysis (PCA) of observed water parameters at Cowley Islet (CoI), Roca Redonda (RoR), and Bolivar Channel (BoC).** DO: dissolved oxygen; Chl a: chlorophyll a; $PO_4^{3-}$, phosphate; $NO_x^-$, nitrate/nitrite; DOC, dissolved organic carbon; TEP, transparent exopolymers; POC, particulate organic carbon.

CoI with median values of 6.78 mg $L^{-1}$, and lowest at BoC with 6.43 mg $L^{-1}$. Inorganic nutrients, chlorophyll a, and POC exhibited similar patterns between sampling sites with values 1.5 to 2 times as high at CoI and BoC than at RoR (Table 1). For instance, while median POC concentrations of 32.93 and 36.65 mg $L^{-1}$ were measured at CoI and BoC, respectively, POC concentrations at RoR did not exceed 22.37 mg $L^{-1}$. Conversely, we observed highest DOC concentrations at RoR and BoC, with median values of 100.54 μmol $L^{-1}$ and 92.14 μmol $L^{-1}$, respectively, as opposed to 80.87 μmol $L^{-1}$ at CoI. The amount of TEP in the water column did not differ significantly between sampling sites (Table 1). Additional to these patterns among sampling sites, most of the measured parameters, except pH, were quite heterogeneous even at the same site, so that the within-site variation was sometimes as large as the between-site variation (Table 1).

The trends in water and carbon pool parameters were summarized in a PCA, which showed well-defined differences among sampling sites (Fig. 2). RoR was separated from CoI

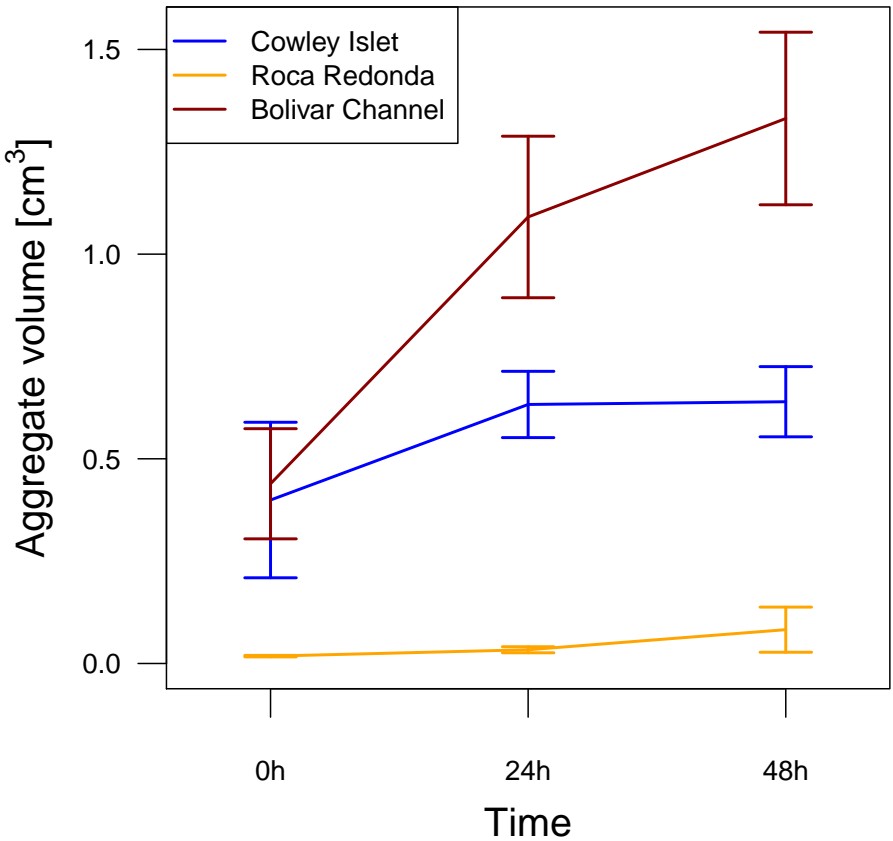

**Figure 3** **Aggregation potential at Cowley Islet (CoI), Roca Redonda (RoR), and Bolivar Channel (BoC) based on rolling tank experiment.** The mean of total aggregate volume per rolling tank ±standard error is shown for each day during the experiment. Sample sizes: 5 (CoI), 4 (RoR), 5 (BoC).

and BoC along principal component (PC) 1, which captured 41.89% of the total variation in the data set. This separation was driven mostly by pH, nitrate/nitrite, chlorophyll a, and POC. Additionally, CoI was separated from RoR and BoC along PC2, which captured 27.59% of the total variation in the data set and was mostly determined by temperature, salinity and DOC (Fig. 2).

## Aggregate formation

After filling the rolling tanks, the total aggregate volume per rolling tank at CoI and BoC of $0.40 \pm 0.19$ and $0.44 \pm 0.13$ cm$^3$ (mean $\pm$ standard deviation), respectively, was about one order of magnitude higher than at RoR (Fig. 3). Over the course of the experiment, aggregate volume in the rolling tanks increased at all sites (GLMM, $F_{2,37} = 6.26$, $p = 0.005$). This potential for aggregate formation differed significantly between sampling sites (GLMM, $F_{2,37} = 25.29$, $p < 0.001$). The strongest increase was observed at BoC, where total aggregate volume had more than tripled after 48 h ($1.33 \pm 0.21$ cm$^3$) compared to the beginning of the experiment (Fig. 3). At CoI total aggregate volume increased only slightly to $0.64 \pm 0.09$ cm$^3$, whereas at RoR even after 48 h total aggregate volume ($0.08 \pm 0.06$ cm$^3$) did not reach

initial mean values reported for the other two sites (Fig. 3). Additional to the differences in total aggregate volume between the sampling sites, the size distribution of aggregates was markedly different, with larger aggregates in the range of 5 to 10 mm dominating the rolling tanks at CoI and BoC compared to RoR, where the size of the majority of the aggregates did not exceed 3 mm (Fig. S2). Therefore, even though there were more, but smaller, aggregates in the rolling tanks at RoR, their total volume was not comparable to that of the fewer, but larger, aggregates at CoI and BoC.

## Bacterial diversity and community composition

A total of 798,402 quality-filtered, non-singleton, and taxonomically classified sequences were obtained, which clustered into 14,811 OTUs. After the merging of technical replicates and removal of OTUs not occurring in at least two technical replicates per sub-site, 90% of the sequences were retained while denoising the data set by excluding 60% of the OTUs, resulting in a final number of 717,420 sequences and 5,970 OTUs.

Effective species richness (inverse Simpson index) ranged from a minimum of 2.31 in the FL fraction at RoR to a maximum of 65.62 in the PA fraction at CoI. There was no consistent difference between FL and PA effective species richness within the same water sample (paired Wilcoxon test, $V = 47$, $p = 0.761$), although values varied considerably (Table 1). Within both the FL and PA fraction, highest median inverse Simpson indices of 33.82 and 48.12, respectively, were recorded at CoI, followed by 21.04 and 17.80 at RoR and 8.46 and 4.85 at BoC, where effective species richness was within a similar range (Table 1).

At class-level, the bacterial communities around the Galapagos Archipelago were dominated by *Gammaproteobacteria* (48% of the total number of sequences), *Flavobacteriia* (17%), *Alphaproteobacteria* (16%), and *Cyanobacteria* (10%). Among these dominant bacterial classes, we detected higher proportions of *Flavobacteriia*, *Alphaproteobacteria*, and *Cyanobacteria* at CoI, whereas *Gammaproteobacteria* were predominantly found at RoR and BoC (Fig. 4). At OTU level, pairwise Bray–Curtis dissimilarities within each water sample between FL and PA bacterial communities ranged from 0.40 to 0.88, and were comparable to Bray–Curtis dissimilarities among samples within the same size fraction. At CoI these differences between FL and PA bacterial communities were attributed mostly to a lower proportion of OTUs affiliated with the cyanobacterial genus *Synechococcus* and the OM60 (NOR5) clade of the *Gammaproteobacteria*, as well as a higher proportion of an OTU affiliated with the verrucomicrobial genus *Roseibacillus* in the PA fraction. At RoR and BoC, other OTUs contributed most to the differences between size fractions. There, the PA fraction was often dominated by one OTU affiliated with the gammaproteobacterial genus *Alteromonas*, which constituted 30 to 55% of the sequences in more than half of the samples at RoR and BoC.

Cluster analysis based on Bray–Curtis dissimilarities revealed prominent patterns in bacterial community composition among size fractions and sampling sites (Fig. 4), which we further explored using ANOSIM. In the FL fraction, bacterial communities from CoI were very similar to each other with an average within-group dissimilarity of 0.26, and well-separated from those at RoR and BoC, despite their comparatively low between-group dissimilarity (Table 2). FL communities at RoR and BoC were more heterogeneous and did

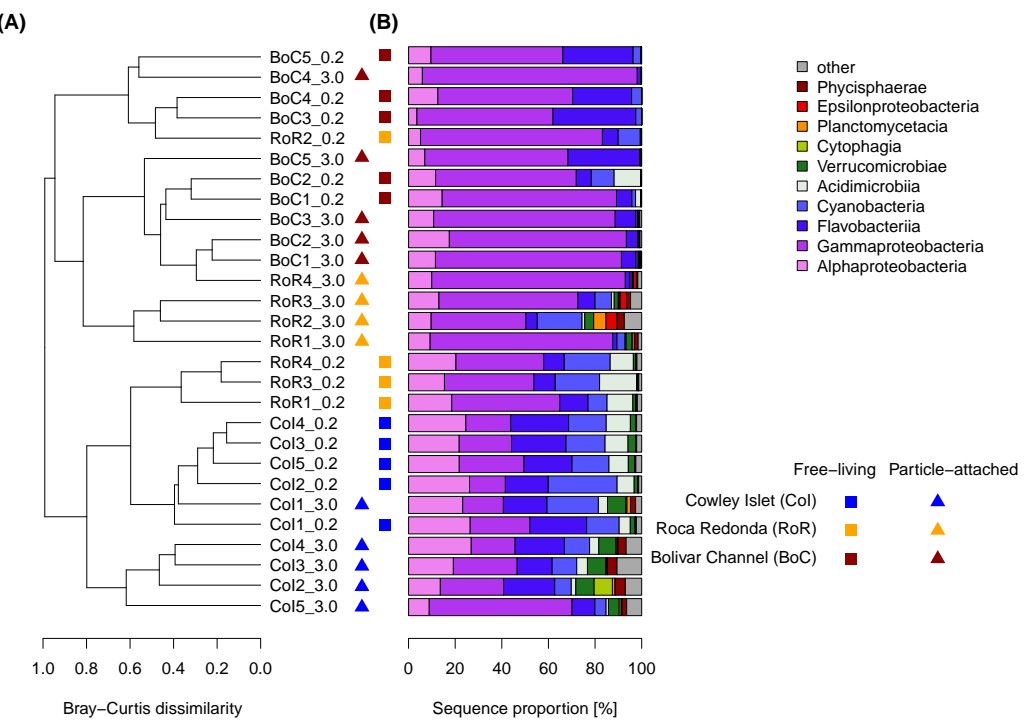

**Figure 4  Composition of the free-living and particle-attached bacterial communities in the water column.** (A) Cluster diagram based on complete linkage hierarchical clustering of Bray–Curtis dissimilarity coefficients. Sample names specify replicate number at each of the three sampling sites: Cowley Islet (CoI1-5), Roca Redonda (RoR1-4), and Bolivar Channel (BoC1-5), and size fraction: free-living ($>0.2$ μ$m$), particle-attached ($>3.0$ μ$m$). (B) Taxonomic composition of the bacterial communities at class level.

not show a strong separation. Within the PA fraction, we observed average within-group Bray–Curtis dissimilarities of approximately 0.50 at all sampling sites. Similar to the FL fraction, strongest differences in bacterial community composition were detected between CoI and each of the other two sampling sites. Additionally, the separation of bacterial communities from RoR and BoC was more pronounced in the PA fraction (Table 2).

Focusing on common and dominant OTUs, RDA showed that sampling site was able to explain 57% and 49% of the variation in bacterial community composition in the FL and PA fraction, respectively (Table 3). The explanatory power of temperature and pH was not as high as that of sampling site, although the resulting RDA ordinations were very similar (Fig. S3). Among the two factors temperature and pH, temperature was able to explain a larger proportion of the variation in community composition than pH. For FL communities, the amount of variation explained by pH was negligible, whereas its contribution to explaining variation in PA communities was more substantial (Table 3). Including additional water and carbon pool parameters did not increase the model fit according to the Akaike Information Criterion (data not shown).
**Table 2  Bray Curtis dissimilarity (BC) of bacterioplankton communities among sampling sites.** Average within and between-group BC and analysis of similarity (ANOSIM) between sampling sites of free-living and particle-attached bacterial communities in the water column.

|  | Within-group BC | Comparison | Between-group BC | ANOSIM R | P-value[a] |
|---|---|---|---|---|---|
| Free-living | 0.67 |  |  | 0.63 | **0.001** |
| CoI | 0.26 | CoI–RoR | 0.61 | 0.72 | **0.017** |
| RoR | 0.55 | CoI–BoC | 0.89 | 0.89 | **0.017** |
| BoC | 0.64 | RoR–BoC | 0.71 | 0.29 | 0.108 |
| Particle-attached | 0.70 |  |  | 0.83 | **0.001** |
| CoI | 0.50 | CoI–RoR | 0.75 | 0.96 | **0.015** |
| RoR | 0.55 | CoI–BoC | 0.91 | 1.00 | **0.015** |
| BoC | 0.49 | RoR–BoC | 0.61 | 0.48 | **0.019** |

Notes.

[a] For pairwise comparisons between the individual sampling sites, Cowley Islet (CoI), Roca Redonda (RoR), and Bolivar Channel (BoC), false discovery rate (FDR)-adjusted p-values are shown. P-values defined as significant at a threshold of 0.05 are bolded.

**Table 3  Contribution of sampling site or temperature and pH to explaining variation in the composition of free-living and particle-attached bacterioplankton communities based on redundancy analysis (RDA).** Variation partitioning was used to test pure effects accounting for the variation explained by the other factors in the model. Akaike Information Criterion (AIC) and adjusted R squared are provided as goodness-of-fit metrics.

| Explanatory variable | AIC | Adjusted R squared | F | df [a] | P-value[b] |
|---|---|---|---|---|---|
| Free-living |  |  |  |  |  |
| Sampling site | 85.31 | 0.571 | 9.65 | 2,11 | **0.001** |
| Temperature + pH | 91.20 | 0.347 | 4.45 | 2,11 | **0.006** |
| Temperature (pure) |  | 0.348 | 7.40 | 1,11 | **0.004** |
| pH (pure) |  | 0.032 | 1.58 | 1,11 | 0.187 |
| Particle-attached |  |  |  |  |  |
| Sampling site | 84.82 | 0.489 | 7.21 | 2,11 | **0.001** |
| Temperature + pH | 89.37 | 0.292 | 3.69 | 2,11 | **0.003** |
| Temperature (pure) |  | 0.213 | 4.61 | 1,11 | **0.008** |
| pH (pure) |  | 0.100 | 2.70 | 1,11 | **0.015** |

Notes.

[a] degrees of freedom (numerator, denominator).

[b] P-values defined as significant at a threshold of 0.05 are bolded.

To further identify, which OTUs were responsible for the patterns in community composition, the differences in the proportion of individual OTUs between sampling sites were tested with ALDEx2. In total, 36 and 30 OTUs were detected as differentially abundant between sampling sites within the FL and PA fraction, respectively (Fig. 5). Of those OTUs, 17 showed similar trends in both size fractions. The majority of differentially abundant OTUs in both size fractions was affiliated with *Gammaproteobacteria*, *Flavobacteriia*, *Alphaproteobacteria*, and *Verrucomicrobiae*, although within each size fraction often different OTUs of those classes exhibited trends between sampling sites. In the FL fraction, differentially abundant OTUs were further found among the *Cyanobacteria* and *Acidimicrobiia*, and in the PA fraction among the *Cytophaga*. In the cases where more

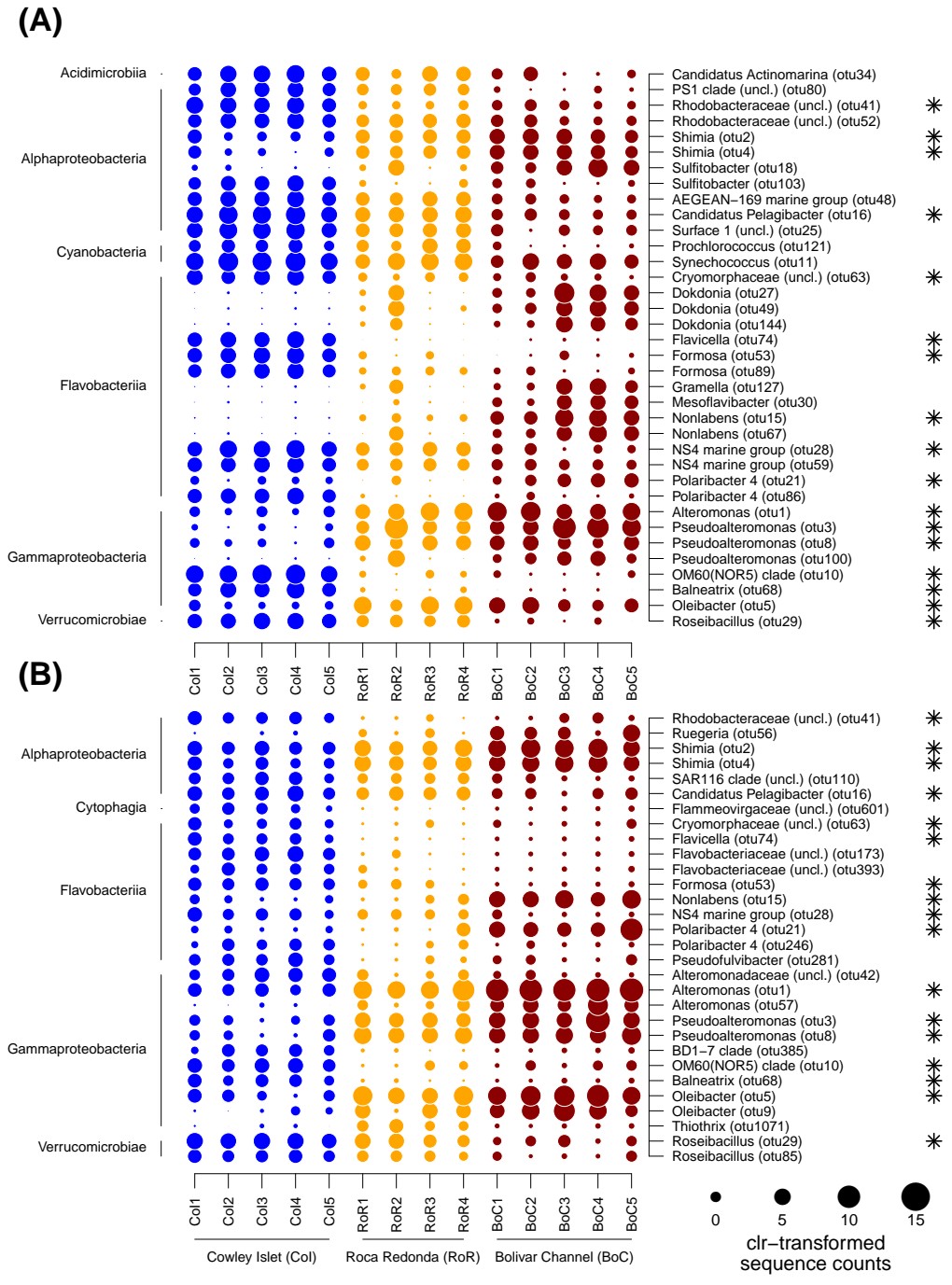

**Figure 5** **Dotplot of differentially abundant OTUs among sampling sites.** (A) Free-living bacterial communities. (B) Particle-attached bacterial communities. Sample names specify replicate number at each of the three sampling sites: Cowley Islet (CoI1-5), Roca Redonda (RoR1-4), and Bolivar Channel (BoC1-5). The size of each dot represents centered log ratio (clr)-transformed sequences counts. Values higher than zero indicate enrichment compared to the other OTUs per sample. The taxonomic affiliation of each OTUs is provided on class (left side) and genus level (right side). Asterisks indicate OTUs detected as differentially abundant in both size fractions.

than one OTU per class was detected as differentially abundant, those OTUs often exhibited divergent patterns between sampling sites (Fig. 5).

Among the *Gammaproteobacteria*, OTUs affiliated with the genera *Alteromonas*, *Pseudoalteromonas* and *Oleibacter* were enriched in both size fractions at RoR and BoC, whereas OTUs affiliated with *Balneatrix* and the OM60 (NOR5) clade were depleted at these sites compared to CoI. Within the PA fraction, one OTU of the genus *Thiothrix* was further only enriched at RoR. OTUs of the flavobacterial genera *Flavicella*, *Formosa*, and the NS4 marine group were characteristic for CoI in both size fractions. OTUs of the genus *Polaribacter* showed divergent trends among sampling sites with different OTUs enriched at either CoI or BoC. Other OTUs of the *Flavobacteriia* that were strongly enriched at BoC were affiliated with *Nonlabens*, *Mesoflavibacter*, and *Dokdonia*, although differences in the latter were only detected in the FL fraction. Among the *Alphaproteobacteria*, more OTUs exhibited differences in their sequence proportions between sampling sites in the FL than in the PA fraction. Especially OTUs of the *Rhodobacterales*, i.e., *Candidatus* Pelagibacter and another representative of the *Surface 1* clade of SAR11, as well as *Sulfitobacter* and other *Rhodobacteraceae*, tended to be higher enriched at CoI than at RoR and BoC, although for some OTUs this trend was restricted to the FL fraction. In the case of *Sulfitobacter*, divergent trends were observed with different OTUs enriched at either CoI or BoC. OTUs predominantly found at RoR and BoC were affiliated with the genus *Shimia* and *Ruegeria*, likewise members of the *Rhodobacteraceae*, although for the latter this pattern was only detected in the PA fraction. Among the *Verrucomicrobiae*, OTUs of the genus *Roseibacillus* were depleted at BoC in both site fractions. OTUs affiliated with *Cyanobacteria* only differed among sampling sites in the FL fraction, with a decreasing trend of two OTUs of the genera *Prochlorococcus* and *Synechococcus*, respectively, from CoI to RoR and BoC. In general, 20 to 80% of the sequences per sample belonged to differentially abundant OTUs. Especially in the PA fraction, there was an overall increase in the sequence proportion of differentially abundant OTUs from CoI to RoR and BoC.

# DISCUSSION

## Physico-chemical characteristics

We characterized the physico-chemical properties of the water column at each of the three sites at BoC, RoR and CoI. As expected, the three sites exhibited different temperature and pH conditions. Additionally, we observed marked differences in inorganic nutrients and chlorophyll a concentrations. At CoI to the east of Isabela Island, temperature and pH were comparable to previous observations at this site, and can be considered representative for the eastern region of the Galapagos Archipelago during that time of the year (*Sweet et al., 2007*; *Liu et al., 2014*). Chlorophyll a concentrations were comparatively low considering the amount of nitrate/nitrite in the water column, resembling HNLC conditions common to the eastern and central equatorial Pacific (*Schaeffer et al., 2008*).

BoC to the west of Isabela Island and CoI to the east, are influenced by different prevailing ocean currents (*Schaeffer et al., 2008*; *Liu et al., 2014*). During upwelling, which constitutes regular conditions at BoC, temperatures drop to about 18 °C, and inorganic nutrient and

chlorophyll a concentrations are expected to be considerably higher than any observed in this study (*Liu et al., 2014*). However, the El Niño event during the sampling period was not as strong as previous events, where temperatures of approximately 26 °C were reached (*Chavez et al., 1999*; *Palacios, 2004*). This suggests that although the upwelling of the EUC was severely reduced, it was not completely inhibited at the time of sampling, thereby limiting but not eliminating the physical resupply of nutrients to surface waters. Probably as a consequence of the weak El Niño, apart from temperature, most of the other observed water parameters were comparable to CoI, and did not resemble the extremes recorded during strong El Niño or La Niña events (*Chavez et al., 1999*; *Palacios, 2004*; *Liu et al., 2014*).

At the submarine volcano at RoR, pH values were markedly lower than what is considered ambient surface ocean pH around the Galapagos Archipelago, or indeed most other oceanic regions today (*IPCC, 2013*), and water temperatures were comparable to BoC. The current system that is influencing BoC, also supplies the north of Isabela Island with nutrient-rich waters under upwelling conditions (*Palacios, 2002*). However, the weak El Niño event at the time of sampling seemed to strongly diminish this influence, thereby resulting in a stronger influence of the SEC with reduced inorganic nutrients and chlorophyll a concentrations, as well as elevated temperatures at RoR (*Schaeffer et al., 2008*; *Liu et al., 2014*).

## Organic carbon pools under differing environmental regimes

The amount of organic carbon, partitioned into particulate and dissolved organic carbon that we measured in the surface water off the Galapagos Archipelago was within the range of previous estimates for the equatorial Pacific Ocean (*Ogawa & Tanoue, 2003*; *Gardner, Mishonov & Richardson, 2006*). Additional to DOC and POC concentration, we evaluated TEP content and aggregation potential to gain a better understanding of the processes that govern particle formation and carbon export. Our results suggest marked differences in the physical and biological processes that influence organic carbon pools at the three sampling sites at the time of sampling, despite a pronounced local variability.

At CoI and BoC, congruently elevated inorganic nutrient, chlorophyll a and POC concentrations may indicate increased phytoplankton abundance, with primary production contributing to the generation of POC (*Ducklow et al., 2001*). However, despite these similarities, DOC concentrations and aggregation potential were both considerably higher at BoC. We observed a positive correlation between DOC concentrations and temperature, suggesting that DOC accumulation may be stimulated by the elevated temperature at BoC, a relationship, which has been previously experimentally documented, although for larger temperature differences than those observed here (*Wohlers et al., 2009*; *Kim et al., 2011*; *Biermann, Engel & Riebesell, 2014*). It has also been shown that increased temperatures promote aggregate formation in phytoplankton blooms due to an increased release of TEP, which is then quickly converted to aggregates (*Claquin et al., 2008*; *Piontek et al., 2009*; *Biermann, Engel & Riebesell, 2014*), and which may explain the high aggregation potential at BoC. Despite the elevated temperatures, the observed patterns in organic carbon pools and aggregate formation at BoC therefore further corroborate the occurrence of a weak El

Niño event at the time of sampling, where some of the upwelling characteristics were still retained and lead to the formation of a phytoplankton bloom.

DOC concentrations at RoR were similarly elevated as at BoC, presumably likewise related to increased temperatures (*Wohlers et al., 2009*; *Kim et al., 2011*; *Biermann, Engel & Riebesell, 2014*) rather than reduced pH (*Zark, Riebesell & Dittmar, 2015*). Interestingly, the aggregation potential at RoR was severely diminished compared to the other two sites despite comparable TEP concentrations. Aggregate formation from TEP can be influenced by a decrease in pH, which reduces the stickiness of TEP and thereby its potential to form aggregates (*Mari, 2008*; *Cárdenas et al., 2015*). The increased $CO_2$ concentrations in the water column at RoR therefore may have directly impacted aggregate formation. Theoretically, the increased availability of $CO_2$ at RoR can also fuel planktonic primary production (*Riebesell et al., 2007*). However, chlorophyll a concentrations recorded at RoR suggest otherwise, indicating that other factors may be limiting phytoplankton growth at this site. Indeed, we detected only low amounts of nitrate and phosphate in the water column. Additionally, unobserved parameters, such as iron, may further limit primary production (*Palacios, 2002*; *Tyrrell et al., 2005*). These circumstances highlight the fact that the strength of $CO_2$ effects is strongly dependent on the environmental setting (*Fu et al., 2008*; *Sala et al., 2016*).

There are several other factors, which may contribute to explaining the observed patterns in organic carbon pools, and which were not investigated here. For instance, the source and composition of the DOC in the water column may affect its fate in the marine carbon cycle (*Thornton, 2014*). Apart from the recalcitrant DOC pool in the water column, processes such as cell lysis, leakage, and exudation by planktonic and benthic organisms replenish the pool of labile DOC (*Haas & Wild, 2010*; *Moran et al., 2016*). For instance, it is likely that at the time of sampling distinct phytoplankton communities occurred at each of the sampling sites, with a more diverse community at CoI than at the other two sites (Fig. S4), which could have influenced the concentration and composition of the organic carbon pool (*Thornton, 2014*). Furthermore, the composition of the benthic community most likely differed among the sampling sites, which fall into separate biogeographic regions with distinct benthic communities (*Edgar et al., 2004*), presumably resulting in differences in the amount and composition of the DOC being released into the water column (*Haas & Wild, 2010*; *Haas et al., 2011*; *Cárdenas et al., 2015*).

## Bacterioplankton communities under contrasting environmental conditions

Bacterial communities in the water column, either free-living or attached to particles, play an important role in the marine carbon cycle, and are strongly interconnected with organic carbon dynamics (*Azam & Malfatti, 2007*). This study is the first to describe the diversity and composition of bacterial communities in the waters around the Galapagos Archipelago, apart from isolated observations during the Global Ocean Sampling (GOS) Expedition (*Rusch et al., 2007*; *Yilmaz et al., 2012*). In general and at a coarse taxonomic resolution, the composition of the bacterial community described in this study is similar to previous observations from tropical ocean surface waters, including those from the vicinity of the

Galapagos Archipelago (*Rusch et al., 2007*; *Zinger et al., 2011*; *Yilmaz et al., 2012*; *Sunagawa et al., 2015*; *Suzuki et al., 2017*). However, at a higher taxonomic as well as spatial resolution, we observed pronounced differences in bacterial community composition between size fractions as well as sampling sites. A sizeable proportion of up to 80% of the sequences per sample were affiliated with taxa that differed in their relative contribution to community composition between the sampling sites, suggesting a major restructuring of bacterial communities with potential implications for bacterial processes involved in carbon cycling in the water column.

Of the observed environmental parameters, temperature was the main determinant of bacterial community composition in both the FL and PA fractions, whereas pH only played a minor role for PA communities. Interestingly, given the correlation of temperature and DOC concentrations in our study, the variation in bacterial community composition explained by temperature, could also be related to changes in DOC concentrations (*Borcard, Legendre & Drapeau, 1992*; *Ramette, 2007*). Changes in bacterial community composition related to temperature and DOC concentrations, respectively, and to a lesser degree pH, further corresponded well with the observed pattern between sampling sites, with communities from RoR and BoC, i.e., the two sites with elevated temperature and DOC concentrations, being more similar to each other than to communities from CoI. In general, these dissimilarities were attributed to a decreased proportion of what could be considered typical oligotrophic bacterioplankton taxa at RoR and BoC, which were strongly dominated by opportunistic copiotrophic bacteria (Table S1). These changes in community composition therefore also affected community evenness, evident in the strongly diminished effective species richness at RoR and BoC. In the remainder of this section we will explore the patterns in bacterioplankton composition in more detail.

The only differentially abundant OTU that was enriched only at RoR was affiliated with the gammaproteobacterial genus *Thiothrix* and occurred almost exclusively in the PA fraction. *Thiothrix* is a filamentous bacterium capable of sulfur oxidation and usually associated with sulfidic hydrothermal vents, where it exhibits an attached lifestyle (*Tarasov et al., 2005*; *Gugliandolo, Italiano & Maugeri, 2006*; *Bailey et al., 2011*; *Giovannelli et al., 2013*; *Zapata-Hernández et al., 2014*). Its occurrence at RoR may indicate the presence of sulfide in the area of the hydrothermal gas venting at RoR. Although there were no obvious indications that sulfide was present at the sampling site, i.e., we did not observe a sulfidic smell or white bacterial mats, it is still possible that the community on the particles originated from sulfidic vents that exist in the vicinity of RoR and host mats of bacterial sulfur oxidizers, or that low concentrations of sulfide persisted (*Gallardo & Espinoza, 2007*). In the first case, the presence of an OTU may also not necessarily implicate metabolic activity (*Lanzen et al., 2011*). Furthermore, we did not detect any OTUs enriched exclusively at RoR in the FL fraction. Presumably, the exposure to the acidified water at RoR was insufficient to trigger changes in the composition of the FL bacterial community (*Weinbauer et al., 2010*).

*Synechococcus* and *Prochlorococcus* are two cyanobacterial genera, which are known to comprise the majority of bacterioplankton primary producers in the oligotrophic eastern equatorial Pacific, especially in HNLC areas (*Partensky, Blanchot & Vaulot, 1999*; *Zubkov*

*et al., 2003*; *Alvain et al., 2008*; *Brown et al., 2014*). Here, they were predominantly found in the FL fraction at CoI, where they may contribute considerably to TEP production (*Iuculano et al., 2017*). At BoC, and to some degree at RoR, they made up a smaller, although in the case of *Synechococcus* still substantial, part of the FL bacterioplankton community. Assuming that a diminished phytoplankton bloom developed at BoC under the weak El Niño conditions, eukaryotic phytoplankton, such as representatives of the genera *Thalassiosira*, *Chaetoceros*, *Emiliania*, which have been previously documented in the Galapagos upwelling system (*Jimenez, 1981*), and which were presumably present at the sampling sites at the time of sampling (Fig. S4), may have dominated primary production instead of taxa better adapted to non-bloom conditions (*Zubkov et al., 2003*; *Brown et al., 2014*).

The remainder of differentially abundant OTUs were affiliated with heterotrophic bacteria involved in the degradation of organic matter. We observed a replacement of organic matter degrading bacterial taxa that may be indicative of functional redundancy, i.e., changing community composition while function is retained (*Allison & Martiny, 2008*), but also of distinct bacterial communities adapted to the degradation of different kinds of organic matter, i.e., performing different functions depending on the concentration and composition of the available organic matter (*Giovannoni, 2005*; *Mou et al., 2008*; *Giebel et al., 2011*; *Moran et al., 2016*). The latter is supported by the differential abundance of mostly oligotrophic taxa at CoI and copiotrophic taxa at RoR and BoC. Among the OTUs predominantly found at CoI, especially members of the SAR11 clade are known to constitute the majority of heterotrophic bacterioplankton in oligotrophic environments, where they are adapted to the degradation of organic matter at low concentrations and predominantly of labile or semi-labile character (*Giovannoni, 2005*; *Moran et al., 2016*; *Bunse & Pinhassi, 2017*). During periods of high organic matter availability, such as phytoplankton blooms, these oligotrophic taxa are generally outcompeted by copiotrophic organisms.

Bacterial communities at RoR and BoC were dominated by OTUs affiliated with representatives of the alphaproteobacterial *Roseobacter* clade (e.g., *Shimia*, *Ruegeria*), several flavobacterial groups, as well as *Alteromonas*, and *Pseudoalteromonas* (*Gammaproteobacteria*). In general, taxa enriched at RoR and BoC have been previously associated with phytoplankton blooms in several oceanic regions, where they contribute to the degradation of mostly labile organic matter during the decline of the bloom (*Buchan et al., 2014*; *Teeling et al., 2016*). Their opportunistic copiotrophic lifestyle is well-suited to exploit short-term organic matter pulses in an otherwise nutrient-poor environment (*López-Pérez et al., 2012*). *Alteromonas* was foremost enriched in PA fraction. Its genetic repertoire encodes for a wide variety of organic matter degrading enzymes, making them highly adapted to utilize the rich availability of substrates on marine aggregates (*López-Pérez et al., 2012*).

The high proportion of *Alteromonas* in the PA fraction also largely contributed to the dissimilarity between size fractions at RoR and BoC, whereas at CoI the prevalence of *Synechococcus* in the FL fraction was most responsible. Therefore, although FL and PA bacterial communities were consistently different from each other (*Rieck et al., 2015*; *Suzuki et al., 2017*), the taxa driving this separation were dependent on the environmental
setting. The differences between FL and PA communities were further determined by changes in OTU proportions rather than by a replacement of dominant OTUs. This observation is consistent with *Bižić-Ionescu et al. (2015)* and *Mestre et al. (2017)*, who detected a considerable proportion of shared taxa between different size fractions, which however differed drastically in their contribution to the community composition of the respective size fractions. Barring technical biases, this situation may indicate that many bacteria are able to switch between FL and PA lifestyles (*Grossart, 2010*; *Bižić-Ionescu et al., 2015*; *Mestre et al., 2017*).

Interestingly, in several instances we observed divergent trends in OTU proportions between the sampling sites within the same taxonomic group, specifically *Polaribacter* (*Flavobacteriia*), *Sulfitobacter* and other *Rhodobacteraceae* (*Alphaproteobacteria*). These OTUs presumably represent different ecotypes of closely related strains that are adapted to contrasting environmental conditions. Such a phenomenon has been previously reported for the aforementioned taxa, where both oligotrophic and copiotrophic strains with opposing genetic potential were discovered within the same genus (*Xing et al., 2015*). Sub-genus and sub-species diversity of ecotypes is widespread in microbial ecology, highlighting the importance of high resolution methods, such as the approach employed here (*Eren et al., 2014*; *Mahé et al., 2014*).

Overall, the taxonomic profile of the bacterial communities at RoR and BoC closely resembled a community that may be expected during and after phytoplankton blooms occurring under upwelling conditions. We consider this further evidence that the west of Isabela Island was experiencing a weak upwelling event, and associated phytoplankton bloom, during the time of sampling. Indeed, considering the temperature and DOC preferences of the taxonomic groups enriched at RoR and BoC, DOC concentrations, presumably together with DOC composition, may constitute the more likely reason for the observed trends in community composition than any of the other measured environmental parameters. Interestingly, bacterial community composition seemed unrelated to POC concentrations, which differed significantly between RoR and BoC. More extensive data on organic matter quality and composition, which may determine community composition as strongly as quantity, as well as bacterial abundance, which may be more strongly affected by organic matter quantity than community composition (*Pinhassi et al., 2004*; *Ortega-Retuerta et al., 2013*; *Buchan et al., 2014*; *López-Pérez et al., 2016*; *Osterholz et al., 2016*), will be necessary to further disentangle organic matter and bacterial community dynamics around the Galapagos Archipelago.

## CONCLUSION

Our study presents a detailed description of environmental conditions, organic carbon pools, aggregation processes, and bacterioplankton composition, highlighting the high heterogeneity of various components of the marine carbon cycle and bacterial communities around the Galapagos Archipelago, and providing baseline data for future investigations. By employing a combination of different analytical approaches, we were able to characterize the conditions at each of the sampling sites more comprehensively than by any method

on its own, and to demonstrate the tight coupling between organic carbon dynamics and bacterioplankton communities. Still, our findings only represent a snapshot in time in a temporally highly variable system. Further research based on repeated observations, including qualitative data on carbon pools, quantitative information on bacterial abundance, as well as estimates of bacterial metabolic activity will be necessary to overcome the limitations and to answer the open questions of the current study, to better understand organic carbon and bacterial community dynamics around the Galapagos Archipelago.

## ACKNOWLEDGEMENTS

We would like to thank the GSC staff and members, especially to Prof. Dr. Steve Walsh and Dr. Carlos Mena for the logistic support and access to the lab facilities. Also, we are very thankful to Maximilian Hirschfeld, Juan García, Daniela Alarcón, and Eduardo Rosero for their assistance and valuable help during the construction of the onboard lab, the field work and sample collection. We would like to thanks the Galapagos National Park, especially Maryuri Yepez and Galo Quezada, for the logistic support. Also, we would like to thank Anny Cardenas for her advice and suggestions. Special thanks to Maximilian Hirschfeld for your comments and input during all the stages of this project, as well as to Ivaylo Kostadinov and the GFBio team for their support during the data submission.

### Funding

This project would not have been possible without the financial support of the Ecuadorian Government through the scholarship program ''Convocatoria abierta 2012'' of the SENESCYT. Further financial support was provided by the Galapagos Science Center (GSC) and the Leibniz Centre for Tropical Marine Research (ZMT). The funders had no role in study design, data collection and analysis, decision to publish, or preparation of the manuscript.

### Grant Disclosures

The following grant information was disclosed by the authors:
Secretaria de Educación Superior, Ciencia, Tecnología e Innovación (SENESCYT).
Galapagos Science Center (GSC).
Leibniz Centre for Tropical Marine Research (ZMT).

### Competing Interests

The authors declare there are no competing interests.

### Author Contributions

- Nataly Carolina Guevara Campoverde conceived and designed the experiments, performed the experiments, authored or reviewed drafts of the paper, approved the final draft.
- Christiane Hassenrück analyzed the data, prepared figures and/or tables, authored or reviewed drafts of the paper, approved the final draft.

- Pier Luigi Buttigieg conceived and designed the experiments, analyzed the data, authored or reviewed drafts of the paper, approved the final draft.
- Astrid Gärdes conceived and designed the experiments, contributed reagents/materials/analysis tools, authored or reviewed drafts of the paper, approved the final draft.

## Field Study Permissions

The following information was supplied relating to field study approvals (i.e., approving body and any reference numbers):

Galapagos National Park issued permit approval 'PNG Permiso de investigacion científica PC-86-14' for sampling sea water. The Environmental Ministry of Ecuador approved the export clearance number 2419373 for filtered sea water samples to be exported to the ZMT, Bremen, Germany. The DNA samples remain stored at the ZMT and will not be used for any purpose, unless with prior consent of the Galapagos National Park and Environmental Ministry of Ecuador.

## DNA Deposition

The following information was supplied regarding the deposition of DNA sequences:

The sequences were deposited in the European Nucleotide Archive (ENA) using the data brokerage service of the German Federation for Biological Data (GFBio), in compliance with the Minimal Information about any (X) Sequence (MIxS) standard and are available under the accession PRJEB27168.

## Data Availability

All associated data (except sequence information) and code are available at Pangaea: Guevara, Nataly; Hassenrück, Christiane; Buttigieg, Pier Luigi; Gärdes, Astrid (2018): Water chemistry and aggregates analysis from the Galapagos Archipelago sampled on Queen Mabel during Galapagos cruise in 2014. PANGAEA, https://doi.org/10.1594/PANGAEA.890858.

## Supplemental Information

Supplemental information for this article can be found online at http://dx.doi.org/10.7717/peerj.5984#supplemental-information.

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
