# Peer review of "Characterization of bacterioplankton communities and quantification of organic carbon pools off the Galapagos Archipelago under contrasting environmental conditions"

_PeerJ, doi:10.7717/peerj.5984_

## Round 0.1 · original submission · Major Revisions

Dear Christiane

I have now received two reviews for your manuscript. You will see that the comments of the reviewers suggest that the manuscript contains some interesting data worthy of publishing. However, both also thought that the presentation of the data and discussion need some considerable amount of work. Thus, there is some clearly good data here and an interesting story that would gain a lot by a revamp in the presentation and structure of the manuscript.

Reviewer 1 ·

Basic reporting

Review Guevara Campoverde et al.

Reference. 28742

Characterization of bacterioplankton communities and quantification of organic carbon pools off the Galapagos Archipelago under contrasting environmental conditions

Summary

The manuscript “Characterization of bacterioplankton communities and quantification of organic carbon pools off the Galapagos Archipelago under contrasting environmental conditions” describes the measurements of physico-chemical data, carbon pools, and microbial community composition at three sites in the Galapagos Islands, which have the potential to become test sites for ocean warming and ocean acidification due to their characteristics. These characteristics were not as prominent as in other years, probably due to an interference by a weak La Niña event. Therefore, no clear sites for the investigation of OW and OA effects were found. Furthermore, the paper describes the abundance of DOC and the formation of POC from waters of this station and finds a microbial community dominated by Gammaproteobacteria (Alteromonas), Roseobacter, and members of the SAR111 clade. The paper determines DOC availability as the driving force for differences in the microbial community, even though temperature is discussed as well as pH.

Experimental design

OK.

Validity of the findings

OK.

Additional comments

General comments

The manuscript is well written and contains several interesting findings. However, the overall structure of the manuscript makes it hard to follow the main hypothesis. It is not clear if the main task was to demonstrate the suitability of the sampling sites to investigate OW and OA or if the microbial community composition of different sites within the Galapagos Islands should be investigated or if the formation of aggregates in response to changing physico-chemical parameters is most important. A connection of the three different topics to form a coherent story is missing. In addition, streamlining all parts would help to increase the quality of the manuscript. Both are specifically true for the discussion.

Furthermore, the difference in temperature seems not to be significant. The max. value of CoI is 22.13 °C, while the min. value of RoR is 22.14°C. Therefore, temperature dependent oxygen solubility is questionable, especially since BoC and CoI show similar DO values, but should be furthest apart. Some papers cited for the argumentation on the dependency of DOC and aggregate formation on temperature state differences of 2-6°C, not 0.8°C as found in this study. Therefore, the argumentation on temperature needs to be more elaborated or the importance of temperature as a factor needs to be reduced. Especially, since Chl a/ phytoplanktom bloom occurrence is later discussed as factors that may have a higher influence than temperature (s. line 647 ff). Considering then the high abundance of DOC loving Gammaproteobacteria, temperature seems not to be the most likely factor shaping the communities. This discussion needs to be more clear and congruent.

Detailed comments

Abstract

Line 40: replace for with to thrive in or abundant in
Line 41: …taxa, such as members of the SAR11 clade, …

Introduction

Line 54: Comma before as well as
Line 97: Comma after nutrients – Consider splitting sentence

Material and Methods

Line 170: Change thought to through
Line 174: Sentence mentioning Germany is obsolete
Line 224: Accession number correct? ENA doesn’t find the accession number.

Results

Line 398: Candidatus Pelagibacter ubique

Discussion

Line 405: Delete extensive
Line 447: the abbreviation for BoC was introduced already
Lines 460-461: The events are named El Niño and La Niña, check throughout the manuscript
Lines 558-570: It seems odd to start and spend 12 lines on a single taxon abundant only on one site, especially when the taxon cannot be correlated to any measured factor. Even though it is a Gammaproteobacterium, I suggest to reduce this two a sentence and discuss it with the other Gammaproteobacteria.
Line 570: Unclear switch to pH.
Line 606: Members of the SAR11-clade
Line 609: SAR11-clade
Line 609-614: Redundant since protorhodopsin was not measured
Line 616: Replace overgrown
Lines 638-640: Why were these methods not used then?
Line 647 ff: This sentence basically undermines all arguments made for temperature before.

·

Basic reporting

no comment

Experimental design

Although the research question is well defined, considering the results and the data interpretation, it might not be the most relevant. I propose some improvements in the "General comments for the author" section.

The rest of the experimental design is fine, within the aims and scope of the journal with rigorous investigation performed and methods described with sufficient details.

Validity of the findings

Conclusions are well stated but may not be appropriate to the results of this study. Please see in the "General comments for the author" where I propose some improvements.

Additional comments

Summary:
The authors investigated patterns of the bacterial community at three sites in the Galapagos Archipelago used as proxies of three different environmental scenarios: a current scenario (reference site: Cowley Islet), an ocean acidification scenario (Roca Redonda), and an ocean warming scenario (Bolivar Channel). Contextual parameters were measured and metabarcoding was used to study the bacterial community from the free-living and the particle-attached fractions at each sites. In parallel, aggregate experiments were conducted to evaluate the volume/number of organic aggregates formed at each site and further estimate their contribution to patterns of the bacterial community.


Major comments:
Overall, I have mixed feelings about this work. The sampling, the original approach, the experimental procedure and analyses are very sound. However, I feel that the manuscript should be written differently to better highlight the findings and the hard work put in that study. I would advise the authors to refocus the hypotheses of their study without testing the OA and OW effects (even though it was the starting hypothesis). For instance, findings could be better highlighted with hypotheses comparing 3 sites with distinct environmental conditions in a tropical zone with a special focus on the impact of the organic carbon pools. In this example, it would be like testing the effect of three distinct ecological niches on the microbial community. In this case, OA and OW may then be discussed later on in the manuscript as perspectives. Also, this new focus fits better the current title.

Also, the authors may either be more positive in the interpretation of their results (by trimming some of the many “may”, “seem”, “likely”) or just remove assumptions where they are too unsure about them (e.g. L656 “may seem the more likely”).


Specific comments:
Please correct through the manuscript: replace “*on the* class level” by “*at the* class level”.

1. Introduction
L86: There is a typing mistake: subject*ed*“”

2. Materials and Methods
L170: There is a typing mistake: “th*r*ough”
L253-256: Could you explain why the data set was reduced following this approach?

3. Results
L318-320: The sentence is a bit hard to follow, please rephrase.
L343: There is a typing mistake: “most *to* the”
L345: Either specify which samples or the number of samples.
L361: SI Fig 2. Is not an RDA, it is the fig. regarding aggregate sizes.
L394: Instead of “likewise of the Rhodobacteraceae”, the authors may write “as for the Rhodobacteraceae”.

4. Discussion
L461: “[…] did not resemble the extremes recorded during strong El Nino or La Nina events.” Do you have a reference for this statement?
L478: What do you mean by “moderately positive correlation”? Is it a non-significant positive correlation?
L517 & Fig. S4: Where do these phytoplankton data come from?
L523: There is a typing mistake: “within *in*”, *in* to remove.
L555-556: Please rephrase the sentence.
L558: What do you mean by “preferentially found only”?

Paragraph starting L577: This part is a bit confusing as the authors still use the OW and OA hypotheses to discuss cyanobacterial patterns although they were falsified in the 4.1. and 4.2. sections.

L634: Why specifying the class for Polaribacter and not for Sulfitobacter and the other Rhodobacteraceae?

To further understand the community dynamics and their correlations with the different environmental conditions, the authors may also propose a functional approach in their perspectives. This would make sense as they hypothesize several times that despite some community turn over, the community may present some functional redundancy.

Figures
Fig. 2 : This is minor but the reading on the PCA may be nicer if the names of the parameters and samples were not overlapping.
Fig. 4: Please correct: “Taxonomic composition of the bacterial communities *at the* class level”

Regarding the supplementary figures, the legends were not easy to find, the authors may add a file document with the legends.

Also, I could not find where the Procrustes figure (SI Fig. 3) was discussed in the text.

---

## Round 0.2 · Minor Revisions

Dear Christiane,

Both reviewers have now only minor editorial recommendations. They both think that the manuscript has improved considerably and all looks really good now in my opinion too.

Reviewer 1 ·

Basic reporting

Review Guevara Campoverde et al.

Reference. 28742

Characterization of bacterioplankton communities and quantification of organic carbon pools off the Galapagos Archipelago under contrasting environmental conditions

Summary

The manuscript “Characterization of bacterioplankton communities and quantification of organic carbon pools off the Galapagos Archipelago under contrasting environmental conditions” has improved significantly since the last review process and is now ready for publication with minute comments.

Experimental design

OK

Validity of the findings

OK

Additional comments

General comments

The microbial community composition on aggregates was subject of some studies recently and has been subject of reserach for years, as shown in Bizic-Ionescu et al., 2015; Thiele et al., 2015, and earlier Simon, Grossart, and others. It would be interesting to compare the results in this study to previous studies.

Detailed comments
Material and Methods

Line 159 ff: How did you measure the aggregate sizes? Opening the tank and measuring them according to Iversens methods would have been to invasive to make a claim on size development afterwards.
Discussion

Line 394: Change he to the.

Line 573-577: Rephrase the sentences

·

Basic reporting

OK

Experimental design

OK

Validity of the findings

OK

Additional comments

The authors have well addressed the different points raised by the other reviewer and I. The manuscript now reads very well, I enjoyed reading this new version. I only have some minor comments:

1. In the earlier version of the manuscript, I commented on why the authors reduced their dataset before the RDA (now line 233). Their 4 reasons make sense but this may not be so common so it could help if these reasons were stated in the manuscript.

2. SI figure 4
The authors may mention somewhere in the mansucript the % chloroplasts/ coverage expected with the V34 primers.

3. Also, please make sure the sequences on ENA are available for publication.

4. Specific comments:
L224: Would this read better?: “The relationship *between* alpha diversity *and* temperature and pH”
L298: There is typing mistake: “ex*lc*uding”
L394: There is typing mistake: “**he El Niño »
L429: There is typing mistake: “due to *i*n increased release of TEP”
L520-521: Please rephrase.

---

## Round 0.3 · accepted · Accept

Dear Christiane

Many thanks for dealing with the reviewers suggestions and corrections. It all looks good for me. Best wishes. Alex

#